# Isolation, In Vitro and In Silico Anti-Alzheimer and Anti-Inflammatory Studies on Phytosteroids from Aerial Parts of *Fragaria × ananassa* Duch

**DOI:** 10.3390/biom12101430

**Published:** 2022-10-06

**Authors:** Mater H. Mahnashi, Osama M. Alshehri

**Affiliations:** 1Department of Pharmaceutical Chemistry, College of Pharmacy, Najran University, P.O. Box 1988, Najran 61441, Saudi Arabia; 2Department of Clinical Laboratory Sciences, Faculty of Applied Medical Sciences, Najran University, P.O. Box 1988, Najran 61441, Saudi Arabia

**Keywords:** *Fragaria × ananassa*, phytosteroids, anticholinesterase, COX, 5-LOX and antioxidant

## Abstract

Based on the pharmacological importance of different species of fragaria, this research was carried out for the isolation of bioactive compounds from *Fragaria × ananassa*. Using the conventional gravity column chromatography followed by small analytical column purification, two major components were isolated from the plant materials. The structures of both compounds (**1** and **2**) were accurately confirmed with GC-MS analysis by comparison of the fragmentation pattern within the library of the instrument. Further, the NMR analysis was also used to supplement the structural evidence. Compound **1** was observed to be 4,22-cholestadien-3-one, while compound **2** was identified as stigmast-4-en-3-one. Both compounds were evaluated for anticholinesterase, COX/LOX inhibitions and antioxidant assays. Compound **1** exhibited the IC_50_ values of 20.29, 27.35, 10.70, 80.10 and 7.40 μg/mL against acetylcholinesterase, butyrylcholinesterase, COX-2, COX-1 and 5-LOX, respectively. Similarly, the IC_50_ values of compound **2** against the same targets were 14.51, 10.65, 8.45, 109.40 and 8.71 μg/mL. Similarly, both compounds were less potent in ABTS and DPPH targets with IC_50_ values in the range of 185.83–369.86 μg/mL. Despite the low potencies of these compounds in antioxidant targets, they can be considered as supplementary targets in Alzheimer and inflammation. The molecular docking studies for the in vitro anti-Alzheimer and anti-inflammatory targets were also performed, which showed excellent binding interactions with the respective target proteins. In conclusion, the isolated phytosteroids from *Fragaria × ananassa* were evaluated scientifically for anti-Alzheimer and anti-inflammatory activities using in vitro and molecular docking approaches.

## 1. Introduction

Medicinal plants have been a rich source of potential bioactive compounds [1]. Although the isolation of bioactive compounds is a tedious process compared to synthetic compounds, the bioactive compounds from medicinal plants are supposed to be safer as compared to the synthetic drug products [2]. The major classes of compounds that can be isolated from medicinal plants are phenolics, flavonoids, alkaloids, glycosides, steroids, terpenoids and others. Among these, the phenolics are compounds that are well known as natural antioxidants [3]. Similarly, alkaloids and steroids are famous for their analgesic, anti-pyretic and anti-inflammatory potentials [4]. The plants’ steroids, or phytosteroids, have been previously studied for anti-Alzheimer and anti-inflammatory potentials [5]. Previously, various bioactive compounds have been identified and isolated from different species of fragaria [6]. Specifically, ellagic acid, flavonoids and phenolic-type compounds have been isolated from the *Fragaria × ananassa* [7,8,9,10].

Steroids play a significant role in the treatment of various inflammatory conditions, such as inflammatory bowel diseases (IBDs), cardiovascular diseases, Crohn’s disease, rheumatoid arthritis (RA), multiple sclerosis (MS), hypercholesterolemia and type I diabetes mellitus [11]. Phytosteroids are steroidal compounds of natural products obtained from plant origin that exhibit a wide range of pharmacological and biological activities [12]. Glucocorticoids suppress the COX-2 gene to exhibit their anti-inflammatory action; similarly, phytosteroids also showed their anti-inflammatory action by targeting COX-2 [13]. Phytosterols (a subgroup of steroids), including β-sitosterol, campesterol, stigmasterol, ergosterol and ergosterol acetate, have demonstrated anti-inflammatory activity through several mechanisms, such as by inhibiting the expression of TNF-α, the release of NO and by blocking COX-2 activity, inducible nitric oxide synthase (iNOS) and ERK pathways in LPS-induced macrophages. The higher anti-inflammatory activity among them was shown by β-sitosterol compared to campesterol and stigmasterol [12,13,14]. Phytosteroids obtained from the *Ichnocarpus frutescens* root exhibited anti-inflammatory activity by downregulating COX-2, TNF-α, IL-6, metalloproteinase-13 (MMP-13) and denaturation of protein [15]. Wang et al. studied the anti-inflammatory activity of nine new steroidal saponins isolated from the berries of *Solanum nigrum* and revealed their inhibitory potential in the production of nitric oxide (NO) induced in LPS-stimulated RAW macrophage [16]. Plant steroids, such as withanolide-M and stigmasterol obtained from *Withania somnifera,* exhibited neuroprotective potential against neurological disorders caused by the activation of nitric oxide synthase [17]. Another study reported the neuroprotective effects of the steroidal components of *Withania somnifera* by scavenging free radicals generated during the early stages of neurodegenerative disorders, such as Alzheimer’s disease (AD) [18]. β-sitosterol obtained from *Polygonum hydropiper* L. is a phytosterol on which anti-Alzheimer’s studies have been conducted previously [5]. Apart from this, the other phytosterols whose neuroprotective effects against neurodegenerative disorders have been reported may include stigmasterol, campesterol, lanosterol, brassicasterol, 4,4-dimethyl sterols and 24(S)-saringosterol [19].

Alzheimer’s disease is a common neurological disorder of the aged population. This disease starts slowly with aging and becomes severe with time, eventually leading to memory and cognition loss. The deficiency of vital neurotransmitter acetylcholine in the synaptic region of aged people causes Alzheimer’s disease [20]. There are a number of approaches to combat the disease condition. However, it is not possible to completely cure the patient. To feel the patient better and to restore their daily life, various biochemical targets are known for treatment. Among the biochemical targets, acetylcholinesterase and butyrylcholinesterase play a role in restoring the neurotransmitter deficit region. Therefore, the use of inhibitors of acetyl and butyrylcholinesterases are among the treatment strategies for Alzheimer’s disease. It has also been evident that AD leads to inflammation [21]. The expression of the cyclooxygenase enzyme in the neuronal region causes neurodegeneration. Similarly, lipoxygenase is an enzyme of the central nervous system. It catalyzes arachidonic acid conversion into leukotrienes [22]. The inhibition of both cyclo- and lipoxygenase enzymes protects from inflammation. Therefore, inhibitions of these enzymes, specifically 5-lipoxygenase, protect from neurotoxicity. With the onset of an inflammatory process in the body, there is an increase in the level of free radicals [20]. The human physiology has its own defense system to fight against the excess of free radicals produced due to inflammation. However, the excessive process of inflammation leads to more free radicals, which is difficult to be overcome by the human defense system. Therefore, as a supplementary treatment for Alzheimer’s and inflammation, it is better to provide a suitable antioxidant for the patient.

Strawberry (*Fragaria × ananassa* Duch.) belongs to the family rosaceae and is a shrub producing spherical red berry fruit, which is used for the production of jellies, jams and marmalades and also eaten raw due to its unique flavor and desirable color, taste and texture [23]. Strawberry has a high content of vitamin C, folate and a variety of polyphenolic compounds with strong antioxidant and other pharmacological activities [6]. Several studies have reported the health importance of strawberry and its components in oxidative stress, apoptosis, diabetic nephropathy and in diabetes pathophysiology [6]. The antioxidant activity of strawberry against various free radicals, such as superoxide radicals, hydroxyl radicals, hydrogen peroxide radicals and singlet oxygen, is due to the presence of its high level of phenolic contents [24]. In several in vitro and in vivo studies, the anti-inflammatory and antioxidant properties of strawberries were reported previously [25]. Strawberries contain ellagic acid, which has been reported to exhibit a wide range of biological properties, including cancer prevention, radical scavenging, antibacterial effects and anti-inflammatory properties [26]. Strawberries have neuroprotective effects due to the presence of anthocyanidin, which inhibits proteasome activity [27]. Quercetin and naringenin are the bioactive compounds of strawberry and exhibit neuroprotective properties by inhibiting ROS formation caused by the beta-amyloid protein [28]. Devore and his co-workers demonstrated that long-term consumption of strawberries decreased the rate of decline in cognitive function [29]. Medicinal plants are a major source of phytosterols [30]. Apart from medicinal plants, phytosterols occur abundantly in food based on edible plants [31], nuts [32], olive oil [33] and other vegetable oils [34]. The major focus of the current researchers in the field of medicinal chemistry is to explore new molecules for the vital pharmacological targets [35,36,37]. The goal of new drug molecules can be achieved by a synthesis of potential molecules or a discovery from natural sources [38,39,40]. Both synthetic and natural compounds have their own advantages and disadvantages for the researchers. Synthetic compounds can be easily repeated in the laboratory [41]; on the other hand, the repetition of naturally isolated compounds is quite challenging [42]. The molecular docking approach is used to find out the binding interactions of a compound in a specific target protein [43,44,45]. This is a convenient in silico approach where medicinal chemists can find the possible targets for a new molecule [46]. This approach is quite helpful when it is used in natural product chemistry in which the identified compounds can first be checked by the molecular docking studies. By using the molecular docking studies in this research, it was concluded that AChE, BChE, COX and LOX were suitable targets for our isolated compounds. Based on the literature, the current study aimed to isolate the steroidal components from strawberry leaves and to explore their anti-Alzheimer, anti-inflammatory and antioxidant properties.

## 2. Results

### 2.1. Chemistry of Isolated Compounds

Both compounds had a steroidal unit in common. The common name of compound **1** was confirmed by GC-MS as 4,22-cholestadien-3-one by matching its fragmentation pattern with the standard library. The experimental GC-MS data and fragmentation pattern of compound **1** were in close correlation with the known library data, which were observed as 393 (4%), 367 (19%), 349 (5%), 298 (21%), 271 (37%), 245 (20%), 213 (17%), 173 (18%), 161 (25%), 149 (42%), 109 (16%), 95 (50%), 69 (76%) and 55 (100%). Similarly, compound **2** was 4-stigmasten-3-one, as observed from the GC-MS library comparison. The experimental GC-MS data and fragmentation pattern of compound **2** were correlated with the known library data, which were observed as 397 (5%), 370 (12%), 327 (4%), 289 (17%), 245 (3%), 229 (34%), 213 (5%), 187 (12%), 149 (22%), 124 (100%), 95 (28%), 55 (35%). The chromatograms related to compounds **1** and **2** are provided in Appendix A.

In the ^1^H NMR spectrum of compound **1**, the singlet at 5.73 ppm consisting of one proton represented the alkene proton at position 4, as shown in Figure 1. Similarly, the alkene protons at positions 22 and 23 gave two distinct splitting patterns at 5.34 and 5.46 ppm. A doublet of a triplet with the coupling constant values of 1.77 and 7.68 Hz at 5.34 ppm was elucidated for the proton at position 22. The two diastereotopic protons at position 24 split the doublet of the alkene proton (position 23) into a ddd (doublet of the doublet of the doublet) with the coupling constant values of 1.23, 4.73 and 7.58 Hz. The two methyl groups at positions 18 and 19 appeared at chemical shifts of 0.66 and 1.18 ppm, respectively. The splitting between 0.94 and 1.12 ppm can be attributed majorly to the protons at positions 21, 25, 26 and 27. It was difficult to assign the overlapping multiplets between 1.22 and 2.55 ppm, but these protons gave a typical pattern of steroidal moiety, as published in the literature [47]. Based on the GC-MS analysis, fragmentation pattern, NMR analysis and comparison with the published literature, compound **1** was observed to be 4, 22-cholestadien-3-one. The IUPAC name of compound **1** is “(Z)-10,13-dimethyl-17-(6-methylhept-3-en-2-yl)-1,2,6,7,8,9,10,11,12,13,14,15,16,17-tetradecahydro-3H-cyclopenta[a]phenanthren-3-one”.

The ^1^H NMR of compound **2** was also observed and compared with the structure identified by the GC-MS analysis. It was observed that the ^1^H NMR was in agreement with the given structure. The most distinct splitting was observed at 5.37 ppm. The doublet with a coupling constant value of 5.16 Hz with an integration value of 1 represented the single alkene proton at position 4. The two methyl groups at position 18 and 19 of the steroidal moiety gave two singlets (3H each) at 0.64 and 1.00 ppm. The splitting between 0.70 and 0.94 ppm (15 protons in total) majorly represented the protons at positions 21, 26, 28 and 29. The multiplet between 2.21 and 2.39 ppm with an integration value of 3 represented two protons of position 6, and one might be the axial diastereotopic proton of position 7. A triplet pattern at 2.01 ppm (2H) typically represented the protons at position 2. Similarly, multiplets were observed between 1.03 and 1.86 ppm, with a total of 18 protons, representing a typical pattern of the remaining steroidal moiety of compound **2**. The data were in agreement with the published literature [48]. Based on the GC-MS analysis, fragmentation pattern, NMR analysis and comparison with the published literature, compound **2** was observed to be 4-stigmasten-3-one. The IUPAC name of compound **2** is “17-(5-ethyl-6-methylheptan-2-yl)-10,13-dimethyl-1,2,6,7,8,9,10,11,12,13,14,15,16,17-tetradecahydro-3H-cyclopenta[a]phenanthren-3-one”.

### 2.2. Anticholinesterase Results

The in vitro cholinesterase (AChE and BChE) inhibitory potentials of test compounds (compound **1** and compound **2**) are shown in a concentration-dependent manner in Table 1. The observed IC_50_ values for compounds **1** and **2** were 20.29 and 14.51 µg/mL against AChE in comparison to the standard galantamine (IC_50_ 7.52 µg/mL). Similarly, in the case of BChE, the IC_50_ values were 27.35, 10.65 and 5.53 µg/mL for compound **1**, **2** and galantamine.

### 2.3. Cyclooxygenase-1 and 2 (COX-1 and COX-2) Inhibitory Results

The inhibitory results of our test compounds (compound **1** and compound **2**) against COX-1 and COX-2 enzymes are summarized in Table 2. The test compounds and standard drugs showed a dose-dependent inhibition at different concentrations, ranging from 62.50 to 1000 µg/mL. In this assay, aspirin and celecoxib were used as standard drugs against COX-1 and COX-2 enzymes, respectively. At a concentration of 1000 μg/mL, phytosteroids **1** and **2** exhibited COX-1 percent inhibition of 66.29 ± 0.43% and 57.57 ± 1.18% with an IC_50_ value of 80.10 µg/mL and 109.40 µg/mL, respectively. Aspirin was used as a positive control drug, exhibiting 75.89 ± 0.20% inhibition of COX-1 at a concentration of 1000 μg/mL and an IC_50_ value of 47.08 µg/mL. Similarly, the percent inhibition of compound **1** and compound **2** against the COX-2 enzyme at the concentration of 1000 μg/mL was observed as 83.13 ± 0.80% and 83.17 ± 0.72% with an IC_50_ value of 10.70 µg/mL and 8.45 µg/mL, respectively. In comparison to our test compounds, the positive control drug was celecoxib, whose percent inhibition against the COX-2 enzyme at the concentration of 1000 μg/mL was observed as 95.20 ± 0.15% with an IC_50_ value of 3.22 μg/mL.

### 2.4. 5-Lipoxygenase (5-LOX) Inhibition Assay Results

The 5-lipoxygenase inhibition results of compound **1** and compound **2** at various concentrations, ranging from 62.50 to 1000 µg/mL, are summarized in a concentration-dependent manner in Table 2. At a concentration of 1000 μg/mL, compound **1** and compound **2** exhibited a percent inhibition of 87.63 ± 0.64% and 85.00 ± 0.30% with an IC_50_ value of 7.40 µg/mL and 8.71 µg/mL, respectively. Similarly, the positive control drug was montelukast, exhibiting a 93.55 ± 0.40% inhibition of 5-LOX at a concentration of 1000 μg/mL.

### 2.5. Antioxidant Results

The antioxidant activity of the test compound **1** and compound **2** against ABTS and DPPH free radicals is shown in a dose-dependent manner in Table 3. The percent ABTS inhibition of compound **1** and compound **2** at 1000 μg/mL (highest concentration) was observed as 62.61 ± 0.77% and 65.17 ± 0.72% with an IC_50_ value of 369.86 μg/mL and 185.83 μg/mL, respectively. Ascorbic acid was the positive control, which displayed 79.00 ± 0.16% inhibition at 1000 μg/mL against ABTS free radicals with an IC_50_ value of 21.72 μg/mL. Similarly, compound **1** and compound **2** and the positive control ascorbic acid at the highest concentration of 1000 μg/mL showed 64.79 ± 0.62%, 71.33 ± 0.49% and 84.39 ± 0.60% inhibition against DPPH free radicals with an IC_50_ value of 314.78 μg/mL, 218.83 μg/mL and 11.47 μg/mL, respectively.

### 2.6. Molecular Docking Results

#### 2.6.1. Docking on 1EVE_(AChE)

Compound **1** showed a conventional hydrogen bonding interaction with Gly123. The same compound showed a π-alkyl interaction with Trp279, Tyr334, Phe330, Phe331 and Trp84. Compound **2** showed a conventional hydrogen bonding interaction with Tyr130 and a π-alkyl interaction with Trp84, Trp279, Phe331, Tyr334 and Phe330.

#### 2.6.2. Docking on 4BDS_(BChE)

Compound **1** showed a conventional hydrogen bonding interaction with Trp82 and 430. The same compound showed a π-alkyl interaction with Phe329 and Tyr332, and a π-sigma interaction with Trp430. Compound **2** showed a conventional hydrogen bonding interaction with Trp82 and 430. This compound also showed a π-alkyl interaction with Tyr332. The docking studies are shown in Figure 2, Figure 3, Figure 4, Figure 5 and Figure 6.

#### 2.6.3. Docking on 1EQG (COX-1)

The docking studies of isolated compounds on the COX-1 target were performed as shown in Figure 2. The protein structure from the protein data bank was obtained with code 1EQG, and both compounds were docked into the minimized pocket of the target protein. Compound **A** showed a π-lone pair interaction with Tyr385 and a π-alkyl interaction with Trp387, Phe518 and Tyr355. Compound **B** showed a triple π-alkyl interaction with Tyr355.

#### 2.6.4. Docking on COX 2 Site

Compound **1** showed a π-alkyl interaction with His90, Trp355, Tyr348, Tyr385 and Trp387. Compound **2** showed a conventional hydrogen bonding interaction with His90 and a π-alkyl interaction with Tyr355, Phe518, Trp387, Tyr385, Phe381 and His90.

#### 2.6.5. Docking on 6N2W_(5-LOX)

Compound **1** showed a conventional hydrogen bonding interaction with Arg596 and also showed a π-alkyl interaction His372 and His432. Compound **2** showed a conventional hydrogen bonding interaction with Arg596. The same compound showed a π-alkyl interaction with Trp599, His367 and His372.

## 3. Discussion

Bioactive isolations have been reported from many medicinal plants [49]. The isolated compounds from medicinal plants have been previously reported for the management of Alzheimer’s disease, inflammation and oxidative stress [50,51]. Specifically, previously, several types of compounds have been isolated from *Fragaria × ananassa*. The ellagic acid [7], flavonoids [8] and phenolic compounds [9,10] have been previously reported from *Fragaria × ananassa*. Therefore, from a chemistry point of view, herein, we isolated the steroidal phytocomponents for the first time from this plant.

Phytosteroids and phytosterols have been previously reported from various plant species. Medicinal plants are a major source of phytosteroids and phytosterols [30]. Apart from medicinal plants, they occur abundantly in food based on edible plants [31], nuts [32], olive oil [33] and other vegetable oils [34]. The plant steroids (beta-sitosterol and stigmasterol) have been previously reported for anti-Alzheimer and anti-cancer activities [5,52]. Chemically, our compound **1** was identified as 4,22-cholestadien-3-one. This steroidal moiety and its other close structural analogs have been previously reported. (*E*)-22-cholestadien-3-one has been previously isolated by Zhaohui and co-workers in *Polygala aureocauda* [53]. The *Bidens pilosa* was analyzed by a GC-MS analysis by Shen et al. They identified 138 compounds, including 4,22-Cholestadien-3-one, and performed the anticancer activity analysis [54]. Similarly, 4.97% of 4,22-cholestadien-3-one was identified by a GC-MS analysis of the cylindrospermum [55]. Recently, 4,22-cholestadien-3-one has been isolated from Leucas zeylanica with a targeting protease of SARS-CoV-2 [56]. Therefore, based on the literature, we can say that we isolated 4,22-cholestadien-3-one from *Fragaria × ananassa* for the first time.

Compound **2**, also called Stigmast-4-en-3-one or 4-Stigmasten-3-one or Sitostenone, has a long history in the published literature [57]. The sitostenone has been previously isolated from the fruits of *Rosa laevigata*, exhibiting glucose reuptake and insulin sensitivity potentials [58]. Sitostenone has also been reported from *Leucosidea sericea*, exhibiting anti-inflammatory activity [59]. This compound has also been isolated in the early 1960s from *Quassia amara* [60].

In this research, we initially isolated the compounds from *Fragaria × ananassa*. After isolation, we subjected the two distinct compounds collected to a GC-MS analysis. The GC-MS analysis was based on a comparison of the compounds with those in the Wiley and NIST libraries. Compounds **1** and **2** were identified as 4,22-stigmastadien-3-one and sitrostenone, respectively. After the GC-MS analysis, we also confirmed their structure by using the ^1^H NMR analysis. The two compounds were subjected to in vitro anti-Alzheimer, anti-inflammatory and antioxidant assays. In the anti-Alzheimer assay, we tested our two compounds against AChE and BChE in comparison to the standard galantamine. Our compounds showed potential activity in comparison to the standard drug. Similarly, in the anti-inflammatory assay, the COX-1, COX-2 and 5-LOX enzymes were used. Overall, based on our observations, we can say that the respective standard drugs in the anti-inflammatory assays were only 1.5- to 2.0-fold higher. We also tested a supplementary target of the antioxidant. However, our compounds were comparatively several folds lower in antioxidant activity. A clear reason for this drop in antioxidant activity was the absence of polar hydroxyl groups. We also docked our compounds in the minimized energy pockets of the respective in vitro enzymes. The binding energies observed were also in agreement with our in vitro results.

## 4. Materials and Methods

### 4.1. Plant Samples and Extraction

The fresh plants of *Fragaria × ananassa* Duch., douglas variety, commonly called the edible strawberry, were obtained, and the aerial parts (leaves) were separated. The collected plant materials were washed with fresh water to cleanse them of unwanted material. The plant materials were spread on clean paper and were allowed shade drying for three weeks. After completely drying the plant materials, they were crushed into powder. The powdered plant materials (1.2 Kg) were soaked in a sufficient amount of ethanol and were allowed to macerate for three weeks. Afterward, the crude extract was filtered to obtain the crude ethanolic extract. The plant’s crude extract was concentrated by evaporating the solvent. However, due to the large volume of the plant sample and the solvent, sometimes, the rotary evaporator does not concentrate it completely. To obtain a concentrated-solvent-free crude extract, it was then kept in a water bath for complete removal of the solvent. The final weight of the crude ethanolic extract was 66 g [61].

### 4.2. Isolation of the Steroidal Compounds ***1*** and ***2***

The dried portion of the crude extract (approx. 20 g) was initially loaded on a large silica gel prepacked gravity column. The column length was 76 cm, and the width was 2 cm. The silica gel (high purity, pore size 60 Å for the column) was purchased from the local vendor of Sigma. The slurry of the silica was prepared in n-hexane. The slurry was poured over the top of the column and was gently tapped with a rubber rod until the silica settled down to about 45 cm. Before starting the isolation process, we tested a few solvent systems using the TLC analysis. The precoated TLC plates (silica 60 F_254_ coated on aluminum sheet) were used. Through the TLC analysis, it was confirmed that the resolution of the components was better in the n-hexane and ethyl acetate combination. Therefore, based on the TLC analysis, we chose the n-hexane and ethyl acetate combination as the solvent system for the separation of components. The column was initially started with a 100% non-polar solvent (n-hexane). The polarity of the solvent system changed gradually with the development of the column. The solvent system was partially polarized by adding 5% portion of the polar ethyl acetate solvent. The polarity of the solvent system changed gradually with 5%, i.e., 100, 95:5, 90:10, 85:15, 80:20, 75:25 and 70:30. Thin-layer chromatographic analysis was performed regularly to check the eluted components. The fractions with the possibilities of phytochemicals (as visualized on a TLC plate under UV lamp) were combined and were dried. The semi-purified fraction was further subjected to further purification using a small-size silica gel column. The length of the small column was 30 cm, with 0.5 cm width. The small column was started with pure n-hexane. The column was eluted carefully, and the polarity was gradually changed by adding 2% ethyl acetate. The polarity was slowly changed, and the small fractions (20 mL each) were collected. Routine TLC analysis was performed to check the possibility of the eluted components. At the end of chromatography, compounds **1** and **2** were purified in 245 and 140 mg, respectively. The structures of the isolated compounds were determined by the GC-MS and NMR analyses [47,48].

### 4.3. NMR and GC-MS Analyses

The NMR analysis was performed by using the JEOL ECX400 NMR instrument. The NMR was a 400 MHz instrument. The chemical shift values were observed in ppm downfield to the internal NMR standard TMS (tetramethylsilane). The GC-MS was performed by using the Agilent USB―393752 instrument with HHP 5MS (5% phenylmethylsiloxane capillary column with dimensions of 30 m × 0.25 mm and film thickness 0.25 μm). The instrument was equipped with a mass detector. The energy of the electron impact was 70 eV, working with the same procedure as reported earlier [36,40].

### 4.4. Anticholinesterase Assays

In this assay, acetylcholinesterase (AChE, obtained from Electrophorus electricus) and butyrylcholinesterase (BChE, obtained from equine serum) were used to check the potencies of compound **1** and **2** using Ellman’s assay [62]. This enzyme method relied on the hydrolytic of acetyl cholinethioiodide (ATChI) by acetylcholinesterase and butyrylthiocholine iodide (BTchI) by butyrylcholinesterase. As a result of this breakdown, 5-thio-2-nitrobenoate anions formed. The anion formed a complex with Ellman’s reagent/DTNB and resulted in a yellow product. The absorption was measured with a spectrophotometer, as per the protocols [63]. Following Elman’s assay, the solutions of both the standard drug (galantamine) and test compounds (phytosteroids **1** and **2**) were prepared (from 62.25 μg/mL to 1000 μg/mL). An amount of 0.1 M phosphate buffer (8.0 ± 0.1 pH) solution was prepared, whose final pH was adjusted by using the KOH solution. For the preparation of enzymatic solutions, AChE (518 U/mg) and BChE (7–16 U/mg) were dissolved in phosphate buffer (pH 8.0), which was further diluted to 0.03 U/mL for acetylcholinesterase and 0.01 U/mL for butyrylcholinesterase. Acetyl butyrylthiocholine iodides 0.0005 M and Ellman’s reagent/DTNB 0.0002273 M solutions were produced in distilled water.

In this assay, 50 µL enzyme solution and 50 µL DTNB reagent were added to 1 mL test samples in a cuvette and then incubated at 30 °C for 15 min. Afterward, a 50 µL substrate solution was added to the mixture. At 412 nm, the absorbance was measured for 4 min by using a double-beam UV-visible spectrophotometer. The positive control was galantamine (10 µg/mL), whereas the negative control had all of the above reaction components without the test samples.

### 4.5. Cyclooxygenases (COX 1 and 2) Assays

The Cyclooxygenase enzymes (COX-1 and 2) inhibitory potential of our test compounds was determined according to the previously reported procedure [64]. Initially, different concentrations of the test compounds, ranging from 62.50 to 1000 μg/mL, were prepared. The enzyme solutions with a concentration of 0.7–0.8 μg/10 μL (COX-1) and 300 U/mL (COX-2) were prepared. To start the assay, for enzyme activation, 10 µL enzyme solution (kept in a cool temperature of 4 °C for 5 min) and cofactor containing hematin (50 µL, 1 mM), glutathione (0.9 mM) and TMPD in a Tris-buffer (0.1 M) with a pH 8.0 was added. After that, 60 µL of the enzyme solution was added to 20 µL of the test samples of various concentrations and incubated at room temperature for 5 to 10 min. Then, 20 µL of arachidonic acid (30 mM) was added to start the reaction. The mixture was then again incubated for 15 min at 37 °C, and absorbances were measured at 570 nm with UV-visible spectrophotometer. In this assay, the standard drug was aspirin and celecoxib for COX-1 and COX-2 enzymes, respectively. The % enzyme inhibitions were calculated by using the previously reported formula.

### 4.6. Lipoxygenase (5-LOX) Assay

The lipoxygenase assay on our test compounds was performed by utilizing human recombinant 5-LOX [65]. In this experiment, different concentrations (62.50–1000 μg/mL) of the test compounds were prepared. The enzyme solution with a concentration of 10,000 U/mL was prepared, whereas 80 mM linoleic acid solution was used as a substrate. Solutions were prepared in a phosphate buffer (50 mM) with a pH of 6.3, which was also utilized as the blank. To start, various concentrations of our test samples were dissolved in 0.25 mL of the buffer, to which 0.25 mL enzyme solution was added and incubated at 25 °C for 5 min. Afterward, the substrate solution (0.1 mL) was added to the mixture and shaken well. The absorbance was recorded at 234 nm with UV-visible spectrophotometer. In this assay, the standard drug was montelukast, and using the following formula, the % inhibitions were calculated:$$\%\;enzyme\;inhibition=\frac{X-X1}{X}\times 100$$
where *X* = absorbance of the negative control (with no test sample), and *X*1 = absorbance of the test compound.

### 4.7. DPPH Assay

The antioxidant potentials of isolated compounds (phytosteroids **1** and **2**) were determined following the DPPH standard assay [66]. Initially, DPPH (24 mg) was dissolved in methanol (100 mL). Stock solutions of the test compounds were prepared in different dilutions (62.50–1000 μg/mL). From each dilution, 1 mL was mixed with 1 mL of the DPPH solution. The solution mixtures were incubated at room temperature for 0.5 h. Afterward, absorbance was recorded at 517 nm with a UV-visible spectrophotometer. Ascorbic acid was the standard drug, whereas the DPPH solution without the compound was negative control. The percent radical scavenging potential of the test compounds was calculated using the following formula:$$\%\;DPPH\;activity=\frac{X-X1}{X}\times 100$$
where *X* = absorbance of the control, and *X*1 = absorbance of the test compound.

### 4.8. ABTS Assay

The antioxidant potentials of isolated phytosteroids **1** and **2** were also determined using the ABTS (2,2′-azino-bis(3-ethylbenzothiazoline-6-sulfonic acid)) standard method [67]. In this assay, ABTS (7 mM) and potassium persulfate (2.45 mM) solutions were produced, vigorously mixed and stored in dark for 12–16 h at room temperature. This created free radicals within the sample. Afterward, the addition of 50% methanol and the ABTS radical cation solution were adjusted to pH 7.4 by dilution with the phosphate buffer (pH 7.4). The inhibition potential of compounds **1** and **2** at 62.50–1000 µg/mL concentrations was determined by mixing the test sample (300 μL) with the ABTS solution (3.0 mL) in a cuvette. After 6 min, the decline in absorbance was measured through a double 734 nm beam UV-visible spectrophotometer. The positive control was ascorbic acid, and the assay was repeated in triplicate. The percent radical scavenging potential of the test compounds was calculated using the standard reported formula.

### 4.9. Molecular Docking Studies

In this article, the molecular docking studies on the inhibition of Cyclooxygenase-1 (COX-1), acetylcholinesterase (AChE), butyrylcholinesterase (BChE), 5-lipoxygenase (5-LOX) and Cyclooxygenase-2 (COX2) were shown by the aforementioned, newly isolated compounds **1** and **2** [64,68]. A molecular docking study was performed using the Molecular Operating Environment (MOE 2016.0802) and the BIOVIA Discovery Studio visualizer [69,70,71]. The three-dimensional structure was obtained from the Protein Data Bank (PDB) using the codes: 1EQG (COX-1), 1EVE (AChE), 4BDS (BChE), 6N2W_(5-LOX) and COX2. The best poses were observed to be deep seated into the active site of the target protein (enzyme), showing all the auspicious and major interactions.

## 5. Conclusions

This research work proved scientifically that *Fragaria × ananassa* is a source of phytosteroids. The two isolated steroids (**1** and **2**) were isolated and characterized. The two compounds showed an almost equal pattern of pharmacological activities due to their structural similarities. The anticholinesterase, COX-1/2 and 5-LOX inhibitions showed that both of these phytosteroids are effective in the management of neurological disorder and inflammation. The activities were supplemented by the antioxidant activity, which are useful in both Alzheimer’s and inflammatory targets. The molecular docking studies showed excellent binding energies for AChE, BChE, COX-1/2 and 5-LOX targets. It can be concluded that both of the isolated phytosteroids (**1** and **2**) are effective in in vitro targets of Alzheimer’s and inflammation. This study provides a baseline for the use of these isolated compounds in in vivo and molecular studies.

## Figures and Tables

**Figure 1 biomolecules-12-01430-f001:**
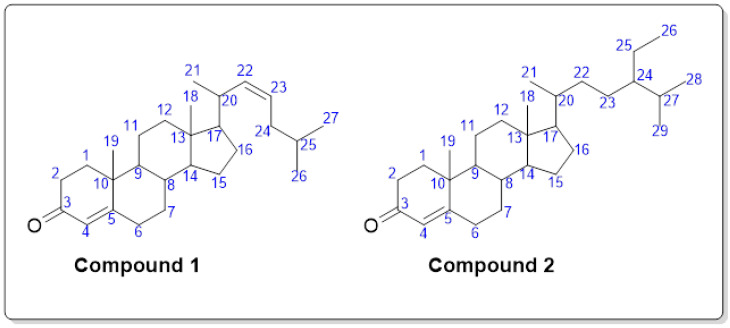
Structure of isolated compounds from *Fragaria × ananassa*.

**Figure 2 biomolecules-12-01430-f002:**
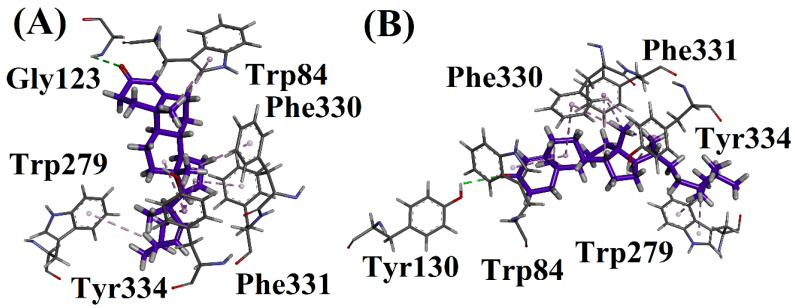
(**A**) Interactions of compound **1** and (**B**) Interactions of compound **2** in active site of 1EVE.

**Figure 3 biomolecules-12-01430-f003:**
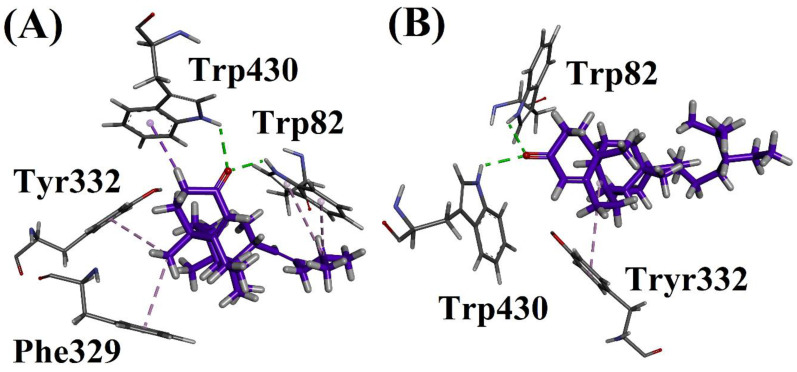
(**A**) Interactions of compound **1** and (**B**) Interactions of compound **2** in active site of 4BDS.

**Figure 4 biomolecules-12-01430-f004:**
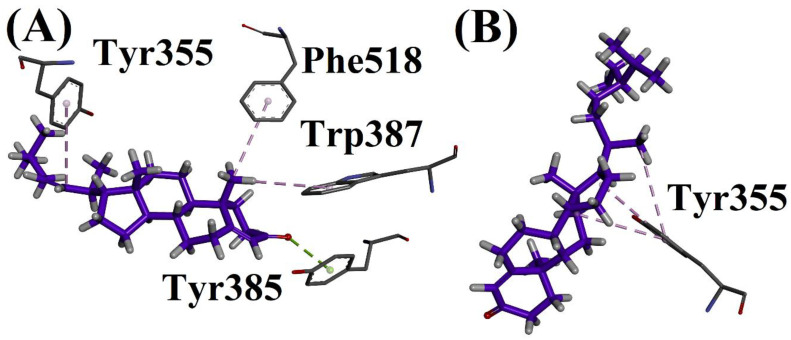
(**A**) Interactions of compound **1** and (**B**) Interactions of compound **2** in active site of 1EQG.

**Figure 5 biomolecules-12-01430-f005:**
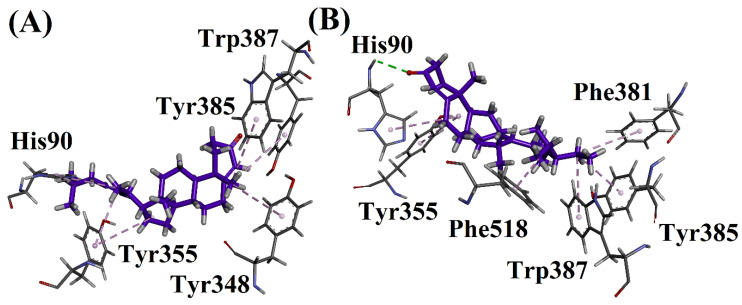
(**A**) Interactions of compound **1** and (**B**) Interactions of compound **2** in COX-2 active site.

**Figure 6 biomolecules-12-01430-f006:**
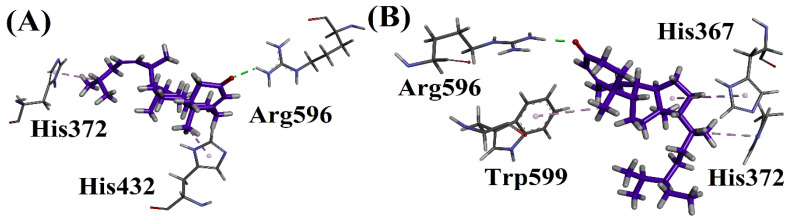
(**A**) Interactions of compound **1** and (**B**) Interactions of compound **2** in 6N2W active site.

**Table 1 biomolecules-12-01430-t001:** Anticholinesterase results of the isolated phytosteroids in comparison to galantamine.

Samples	Concentration	Percent AChE (mean ± SEM)	AChE IC_50_ (μg/mL)	Percent BChE (mean ± SEM)	BChE IC_50_ (μg/mL)
**Compound 1**	1000	85.72 ± 0.79 ^ns^	20.29	84.83 ± 0.62 ***	27.35
	500	77.68 ± 0.63 ***		80.76 ± 0.63 ***	
	250	71.46 ± 0.53 ***		75.70 ± 0.62 ***	
	125	64.78 ± 0.60 ***		66.65 ± 0.78 ***	
	62.5	55.56 ± 0.52 ***		59.81 ± 0.65 ***	
**Compound 2**	1000	80.85 ± 0.18 ***	14.51	83.53 ± 0.20 ***	10.65
	500	75.59 ± 0.30 ***		78.62 ± 0.17 ***	
	250	68.75 ± 0.14 ***		73.42 ± 0.11 ***	
	125	63.47 ± 0.49 ***		66.20 ± 0.15 ***	
	62.5	58.12 ± 0.34 ***		61.35 ± 0.18 ***	
**Galantamine**	1000	87.81 ± 0.60	7.52	89.37 ± 0.64	5.53
	500	82.74 ± 0.61		83.45 ± 0.65	
	250	77.68 ± 0.60		78.37 ± 0.54	
	125	72.63 ± 0.76		74.30 ± 0.61	
	62.5	65.79 ± 0.63		67.42 ± 0.55	

Data are presented as (mean ± S.E.M); two-way ANOVA followed by Bonferroni test were followed. Values significantly different as compared to positive control; n = 3, *** = *p* < 0.001, ns; non-significant.

**Table 2 biomolecules-12-01430-t002:** Cyclo- and lipoxygenase inhibition results of the isolated phytosteroids from *Fragaria ananassa*.

S. No	Conc. (μg/mL)	% COX-2 Inhibition	IC_50_ μg/mL	% COX-1 Inhibition	IC_50_ μg/mL	% 5-LOX Inhibition	IC_50_ μg/mL
**Compound 1**	1000	83.13 ± 0.80 ***	10.70	66.29 ± 0.43 ***	80.10	87.63 ± 0.64 **	7.40
	500	78.83 ± 0.73 ***		59.56±0.45 ***		82.45 ± 0.55 ***	
	250	72.70 ± 0.51 ***		43.54 ± 0.46 ***		76.53 ± 0.41 ***	
	125	66.43 ± 0.70 ***		40.57 ± 0.84 ***		71.42 ± 0.46 ***	
	62.50	61.06 ± 0.70 ***		22.36 ± 0.49 ***		65.68 ± 0.64 ***	
**Compound 2**	1000	83.17 ± 0.72 ***	8.45	57.57 ±1.18 ***	109.40	85.00 ± 0.30 **	8.71
	500	78.30 ± 0.64 ***		51.67 ± 0.11 ***		78.76 ± 0.58 ***	
	250	73.34 ± 0.63 ***		44.86 ± 0.02 ***		73.67 ± 0.61 ***	
	125	68.30 ± 0.64 ***		37.72 ± 0.45 ***		67.74 ± 0.61 ***	
	62.50	61.93 ± 1.13 ***		32.45 ± 0.65 ***		63.47 ± 0.56 ***	
**Celecoxib**	1000	95.20 ± 0.15	3.22	-	-	-	-
	500	91.17 ± 0.53					
	250	86.98 ± 0.85					
	125	81.20 ± 0.65					
	62.50	77.80 ± 0.37					
**Montelukast**	1000	-	-	-	-	93.55 ± 0.40	4.50
	500					89.37 ± 1.65	
	250					85.50 ± 0.40	
	125					79.60 ± 0.90	
	62.50					74.17 ± 0.72	
**Aspirin**	1000	-	-	75.89 ± 0.20	47.08	-	-
	500			71.88 ± 0.20			
	250			66.43 ± 0.29			
	125			59.84 ± 0.32			
	62.50			51.68 ± 0.22			

Data are presented as (mean ± S.E.M); two-way ANOVA followed by Bonferroni test were followed. ** = *p* < 0.01, *** = *p* < 0.001.

**Table 3 biomolecules-12-01430-t003:** Percent DPPH and ABTS activity of isolated compounds.

Samples	Conc (μg/mL)	Percent ABTS Activity (mean ± SEM)	IC_50_ (μg/mL)	Percent DPPH Activity (mean ± SEM)	IC_50_ (μg/mL)
**Compound 1**	1000	62.61 ± 0.77 ***	369.86	64.79 ± 0.62 ***	314.78
	500	54.60 ± 0.80 ***		56.45 ± 0.49 ***	
	250	43.83 ± 0.56 ***		45.75 ± 0.58 ***	
	125	35.69 ± 0.77 ***		37.51 ± 0.77 ***	
	62.5	29.67 ± 0.61 ***		31.53 ± 0.71 ***	
**Compound 2**	1000	65.17 ± 0.72 ***	185.83	71.33 ± 0.49 ***	218.83
	500	57.85 ± 0.97 ***		63.03 ± 0.23 ***	
	250	51.37 ± 1.65 ***		49.00 ± 0.58 ***	
	125	46.73 ± 0.78 ***		42.67 ± 0.89 ***	
	62.5	41.34 ± 1.01 ***		33.00 ± 1.15 ***	
**AA**	1000	79.00 ± 0.16	21.72	84.39 ± 0.60	11.47
	500	74.66 ± 1.20		78.58 ± 0.56	
	250	66.33 ± 0.33		72.29 ± 0.43	
	125	62.50 ± 0.44		66.37 ± 0.58	
	62.5	53.00 ± 0.57		61.30 ± 0.52	

Data are presented as (mean ± S.E.M); two-way ANOVA followed by Bonferroni test were followed. *** = *p* < 0.001.

## Data Availability

The whole data are available within the manuscript and the Appendix A.

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
