# Peer review of "Isolation, In Vitro and In Silico Anti-Alzheimer and Anti-Inflammatory Studies on Phytosteroids from Aerial Parts of Fragaria × ananassa Duch"

_biomolecules, 2022, doi:10.3390/biom12101430_

Round 1

Reviewer 1 Report

Overall the manuscript was not well written.  I have several comments:

1.     1.      The authors should provide the occurrence of phytosterol in their last part of the introduction (line 89-109)

2.      Self citation for reference 20 line 288 was inappropriate. The authors should use the reference regarding the isolation or NMR data of the earlier reports of compounds 1 and 2.

3.      The authors may provide  the chromatogram of GC and the H NMR spectra to check the purity of both compounds

4.      Chemistry part in the Result and Discussion should discuss first the process of structure elucidation and thus the name should come afterwards (line 112-116).

The authors should also provide the trivial name for compound 2.

5.      The authors should provide the NMR data in a Table consisting the NMR data obtained in their experiment in comparison with the earlier reports of the two compounds.

6.      The explanation of the structural elucidation using  H NMR was not clear. The authors should mention the exact signals instead of guessing (Line 127-135).

7.      The Structure of the compounds should be numbered and these numbers can be used  in discussing the NMR data by comparison with the literatures.

8.      The Authors should discuss their chemistry result in comparison with the previous work for the compounds

9.      The later discussion in line 230-259 was not well written and inappropriate. This parts may be suitable in the introduction.

Reviewer 2 Report

This manuscript by Mahnashi and Alshehri describes the isolation of two phytosteroids (compounds 1 and 2) from the leaves of Fragaria x ananassa and their evaluation as anti-inflammatory and anti-oxidant agents, including docking studies with potential targets.

The two isolated steroids are well-known compounds, or at least I presume this. Indeed, the structure reported for compound 1 is actually unprecedented. It is claimed to be cholesta-4,22-dien-3-one; however, the configuration of the double bond in the lateral chain in this compound is E. I believe that the structure has been drawn erroneously with a Z-configured double bond. NMR data might reveal the configuration of the double bond, but they are not reported, nor the spectra in the supplementary files. Compound 2 is also a well-known steroid, whose common name is stigmast-4-en-3-one. Both compounds have been drawn without specifying their absolute configuration; moreover, the usual way for drawing steroids is upside-down with respect to structures in Figure 1.

These two phytosteroids have been previously isolated from a large number of natural sources and their biological activities have been studied diffusely, including as anti-oxidants and cholinesterase inhibitors.

In the Discussion, the Authors state: "The major focus of the current researchers in the field of medicinal chemistry is to explore new molecules for the vital pharmacological targets": these are not new molecules!

Moreover, they state: "The naturally isolated compounds are comparatively safer as compared to the synthetic drugs.". I fully disagree. Generally speaking, the most potent poisons are found among natural products. If the Authors refer to the same compound, the synthesized one will be as safe as the isolated one, at the same level of purity.

For the reasons given above and in consideration of the moderate bioactivities reported, in my opinion this manuscript lacks the level of novelty and interest required for publication in Biomolecules. These results may be more appropriate for a journal specialized in botanical studies.

Reviewer 3 Report

Title

The plant part should be included in the title in particularly as it is not the fruit itself rather the leaves that has been investigated.

Abstract

The identities of the compounds should be given; ‘4,22-cholestadien-3-one’ and ‘stigmast-4-en-3-one

Introduction

L37: Only 6% of plant species have been investigated for their biological activities? Reference 1 was printed in 2006, and this number is for sure not updated.

L50-65: Delete or strongly reduce this paragraph. Lines 54-48 contain enough background on the general potential biological effects.

L78: ..the production of.. rather than ..the of production..

L85: phytosterols rather than phytosterol

Line 89: Strawberry is not evergreen in the northern hemisphere. Moreover, there are several varieties that are not evergreen.

Line 90: Remove ‘dark’

Line 104: include an ‘and’ to give ‘..strawberry, and exhibits..’

The background of the studied assays (anticholinesterase, COX/LOX, antioxidant) is absent/weak.

Results

L118: The short name of cmp 1 is given as 22-cholestadien-3-one. There is however a missing number in this name as the position of the first double bond is missing. In accordance to your molecular structure this should be 4,22-cholestadien-3-one. Also include the shortname of cmp 2.

L141-146: Erase. This information is already included in Table 1.

Table 1: Why is galantamine include? As a reference compound of course (L142), but information is absent in the table-heading. A table is a self-standing information-box in a scientific text, and should include all necessary information to understand the information included in the table.

Table 2: Correct table numbering!!

Discussion

This is introduction-material over again, not a discussion! The section should discuss the results of the presented work in accordance to the literature. An obvious point to start would be with the identities of the isolated compounds. Further, how are the results from the bioassays of the two target molecules compared to the standards? Expectations compared to literature? Error sources?

Material and methods

L262: Which variety or cultivar?

L265: Macerated for 3 weeks?

L269: ‘The obtained crude extract was then kept in water bath for drying.’ Explain.

L273: Dimension of the column? Amount of silica? Type of silica? Conditioning of the column?

What about TLC and the small silica column? Remember that enough information should be given as the reader could repeat the experiments.

L290-297: This should be moved to the introduction section.

L342-351: Normal font, not italic!

L353: Something is missing in this sentence.

L354: What is ABTS?

L354: Do you mean ‘potassium peroxodisulfate’?

L356: ‘Afterwards, the addition of 50% methanol and the ABTS radical cation solution was adjusted by diluting with Phosphate buffer (0.01M, pH 7.4).’ Adjusted to what? Why 50% methanol?

I miss information about GC-MS and NMR??

Conclusion

L376: ‘characterised’ rather than ‘confirmed’.

L379: ‘neurological’ rather than ‘nurological’?

References

There are in general too many references compared to the rather limited scientific contribution of the work. The first author is included in nine references? All from 2021-2022. This is in general not a good indication in science.

Supporting information

The m/z with no further information is normally understood as the molecular mass. However, here it is meant to be the base peak of the spectra. This should be explained. Further, what are the masses of the two compounds? Are the molecular masses detected?

What about NMR spectra?

Round 2

Reviewer 1 Report

The Structure elucidation of compound 1 and 2 was not clear at all and was not convincing. The authors did not provide the chromatogram of GC MS and also the spectra of proton 1H NMR)as requested in the first round. The authors comply in the cover letter that the purity is 99% but they do not provide with the evidence (either from chromatogram of GC MS or the spectra from the proton 1H NMR). They did not provide the data in the manuscript and also in the supporting information.  

In my opinion the structure elucidation of the both compound was not clear. It was not well supported with the fragmentation pattern of the compound and also with the table of NMR data as the molecular weight was not the same as observed in MS spectra.

Line 175-176 : the name of the compound 1 and 2 was not clear

The authors did not provide the table of H NMR data in comparison with the published literature as requested in the first round. The H NMR chemical shift was written well and not consistent. It should use only two digits (for example 2.211 and 2.390 ppm should be written 2.21 and 2.39)

Author Response

Rebuttal and Supp Info attached

Reviewer 2 Report

I have previously reviewed the original version of this manuscript by Mahnashi and Alshehri. No doubts that this revised version has been greatly improved and the Authors have addressed most of the Referees’ critical observations.

Essentially, I think that now the manuscript might be accepted for publication. Nonetheless, in consideration of the low bioactivity found, I still believe that the interest among the readers of Biomolecules will be limited and this paper is more appropriate for a dedicated journal, such as Phytochemistry or Phytochemistry Letters.

Still, the absolute configuration of known compounds 1 and 2 is not depicted in Figure 1. Moreover, I could not find any NMR spectra of the compounds, albeit the Authors claim to have included them in the Supplementary Information.

Author Response

Rebuttal and Supp Info attached

Reviewer 3 Report

There is still no description of the methods used in accordance to GCMS and NMR. This is important!

Author Response

Rebuttal and Supp Info attached
